# Impact of Ar Flow Rates on Micro-Structural Properties of WS_2_ Thin Film by RF Magnetron Sputtering

**DOI:** 10.3390/nano11071635

**Published:** 2021-06-22

**Authors:** Md. Akhtaruzzaman, Md. Shahiduzzaman, Nowshad Amin, Ghulam Muhammad, Mohammad Aminul Islam, Khan Sobayel Bin Rafiq, Kamaruzzaman Sopian

**Affiliations:** 1Solar Energy Research Institute, The National University of Malaysia, Bangi 43600, Malaysia; akhtar@ukm.edu.my (M.A.); ksopian@ukm.edu.my (K.S.); 2Graduate School of Natural Science and Technology, Kanazawa University, Kakuma, Kanazawa 920-1292, Japan; 3Institute of Sustainable Energy, Universiti Tenaga Nasional (@The National Energy University), Jalan Ikram-Uniten, Kajang 43000, Malaysia; 4Department of Computer Engineering, College of Computer and Information Sciences, King Saud University, Riyadh 11451, Saudi Arabia; ghulam@ksu.edu.sa; 5Department of Electrical Engineering, University of Malaya, Jalan Universiti, Kuala Lumpur 50603, Malaysia; aminul.islam@um.edu.my

**Keywords:** tungsten disulfide (WS_2_), thin film, radio frequency magnetron sputtering, gas flow rate, defect engineering

## Abstract

Tungsten disulfide (WS_2_) thin films were deposited on soda-lime glass (SLG) substrates using radio frequency (RF) magnetron sputtering at different Ar flow rates (3 to 7 sccm). The effect of Ar flow rates on the structural, morphology, and electrical properties of the WS_2_ thin films was investigated thoroughly. Structural analysis exhibited that all the as-grown films showed the highest peak at (101) plane corresponds to rhombohedral phase. The crystalline size of the film ranged from 11.2 to 35.6 nm, while dislocation density ranged from 7.8 × 10^14^ to 26.29 × 10^15^ lines/m^2^. All these findings indicate that as-grown WS_2_ films are induced with various degrees of defects, which were visible in the FESEM images. FESEM images also identified the distorted crystallographic structure for all the films except the film deposited at 5 sccm of Ar gas flow rate. EDX analysis found that all the films were having a sulfur deficit and suggested that WS_2_ thin film bears edge defects in its structure. Further, electrical analysis confirms that tailoring of structural defects in WS_2_ thin film can be possible by the varying Ar gas flow rates. All these findings articulate that Ar gas flow rate is one of the important process parameters in RF magnetron sputtering that could affect the morphology, electrical properties, and structural properties of WS_2_ thin film. Finally, the simulation study validates the experimental results and encourages the use of WS_2_ as a buffer layer of CdTe-based solar cells.

## 1. Introduction

Historically, chalcogenides of the first-row transition materials have drawn researchers’ interest for their elementary abundance and impressive structural characteristics [1]. The characteristic features of these chalcogenides are their well-known two-dimensional structures in which metal and chalcogen are periodically arranged. Transition-metal di-chalcogenides (TMDCs) are commonly known as 2D materials that exhibit hexagonal structure, but their hexagonal structure is not atomic-thin like graphene. The overall efficiency of atomically thin TMDCs based devices depends on a number of factors, including stability, thickness, substratum, contacts, temperature, and surface functionalization of material. Few-layered MX_2_ (M = Mo, W; X = S, Se) TMDCs exhibit various fascinating properties associated with their reduced thickness [2,3]. For example, TMDCs undergo a layer-dependent transition in their band structure from an indirect to a direct bandgap semiconductor [4,5]. This transition immediately makes TMDCs attractive for electronics and optoelectronics applications [2,6]. TMDC-based instruments, however, exhibit n- or p-type behavior, contradicting what one would expect from a perfect crystal structure without unsaturated bonds [7].

TMDCs may contain a variety of structural defects in their crystal labels that alter their physicochemical properties significantly. Such observations point to a simple fact: structural defects in TMDCs cannot simply be ignored [8,9]. Tungsten disulfide (WS_2_) exhibits a sandwich-type basic building block structure composed of a sheet of hexagonal close-packed transition-metal atoms between two sheets of hexagonal close-packed chalcogen atoms. Atoms within a sheet are bonded by covalent bonds, whereas the individual sheets are stacked by weaker van der Waals (vdW) bonds. Due to weaker vdW bonds in the latter one, this structure becomes sensitive to any small changes of growth parameter resulting in various types of defects such as stacking defect, edge defect, line defect, etc. [10,11,12,13,14]. Moreover, a small modification of the synthesis path also leads to significant variations in the electronic properties of the material [15]. Based on the intend of application, structural defects can be counterproductive or beneficial [8,16,17], but mostly it has a detrimental effect. In the case of photovoltaic applications, structural defects are mostly harmful. Defects positioned either in the interface of the absorber and window layer or within the absorber layer always hinder the overall performance of solar cells [18]. Realistic incorporation of TMDCs into devices is far from optimal standard. The existence of structural defects in TMDCs is the main bottleneck for which TMDCs are prevented from being integrated into modern devices [19]. Therefore, a comprehensive understanding of structural defects and defect engineering is required.

Radiofrequency (RF) magnetron sputtering is a plasma or ion-assisted deposition process commonly used for the fabrication of thin film due to its high yield rate and ability to produce uniformity on the film [20]. Increasing the working gas flow rate is highly suspected to result in an increase in energy flux at the substrate that can lead to a higher deposition rate and bigger grain growth. Again, with higher gas flow rate and moderately high power leads to a high bombardment of the atom, which may cause sputter damage to the film. Hence, optimization of gas flow rate is necessary to achieve desired material properties [21,22]. Many experimental works have been undertaken to study the effect of process parameters of RF magnetron sputtering on thin film properties. The type of gas and variations in flow rate during sputtering has an enormous effect on thin film properties [23,24,25,26]. Usually, the gas flow rate directly influences the sputtering yield. The energy and energy distribution of sputtered atoms varies with the change of gas flow rate, which influences morphological and electrical properties of the thin film [24,27]. As Ar facilitates to achieve plasma from the source material without taking part in any reaction, it is commonly used for sputtering. Moreover, the gas flow rate has a significant influence on film stoichiometry, phase composition, and preferred orientation as well [28,29,30,31,32]. It is also reported that argon-nitrogen gas mixtures influence the structural properties of the film and improve the adhesivity [33]. However, there have not been many attempts to examine the impact of Ar gas flow rate on the WS_2_ thin film properties. This study is a follow-up work of previously reported studies on WS_2_ thin film [22,23] and devoted to investigating the impact of variation in gas flow rate on the WS_2_ film properties in order to achieve higher crystallite, defect-free, uniform film for the application in the solar cell. As a whole, the objective of this study is to evaluate the micro-structural changes of WS_2_ thin film that occurred due to the variation in gas flow rate.

## 2. Materials and Methods

In this study, WS_2_ thin film was deposited for 30 min on soda-lime glass substrates of 7.5 cm × 2.5 cm × 0.2 cm by RF magnetron sputtering technique. Substrates were cleaned prior to deposition following the sequential cleaning process, such as mechanical scrubbing followed by acetone-methanol-deionized water in ultrasonic bath and later dried with N_2_ gas flow. A 50 mm diameter WS_2_ (99.99%) sputtering target (supplied by Kurt. J. Lesker, USA) was used as the source material. Purging of the sputtering chamber was carried out thrice to remove unwanted particles from the chamber. Afterward, the chamber pressure was brought down to 10^−6^ torr by the turbo-molecular pump. Working pressure ranges from 1.3 to 1.8 Pa with the variation of gas flow rate throughout the deposition process. and substrate temperature was fixed at 100 °C during the deposition. Substrate-to-target distance (sputter up) and substrate holder rotation were fixed at 8 cm and 10 rpm, respectively. In this experiment, Ar gas flow rate was varied from 3 to 7 sccm to identify its impact on the crystallite structure of WS_2_ thin film. To avoid the significant contamination of film by oxygen or hydrogen from the residual atmosphere, initially purging was carried out to the chamber, and later substrates had been preheated by 100 °C for 15 min before deposition. Schematic of sputtering is shown in Figure 1. Deposition parameters used for this study are shown in Table 1. All the samples were kept inside the sputtering chamber upon completion of the deposition until substrate temperature falls to room temperature as a measure to prevent as-grown films from being oxidized.

A BRUKER aXS-D8 Advance CuKα diffractometer (USA) was used to characterize the structural and crystalline properties of the as-grown films. X-ray diffraction (XRD) patterns were recorded in the 2θ range of 10° to 80° with a step size of 0.05° using Cu Kα radiation wavelength, λ = 0.15408 nm. Surface morphology and cross-sectional view of the films were observed by using FEI Quanta 400F (USGS, USA) field emission scanning electron microscope (FESEM) at 10 kV operating voltage. The atomic composition was determined by Horiba EMAX 450 (Horiba, UK) energy-dispersive X-ray spectroscopy (EDX). This method provided the atomic percentage (at %) of W and S atoms present in the sample. The electronic properties of the films were measured by the HMS ECOPIA 3000 Hall Effect measurement system (Bridge Technology, USA) with a magnetic field of 0.57 T and probe current of 100 nA for all the samples. Pinholes were investigated by keeping as-deposited samples over a light source, and the adhesiveness of films was tested by the scotch-tape method.

## 3. Results and Discussions

### 3.1. Structural Analysis from XRD

The XRD analysis was performed to investigate the crystallographic properties of as-grown WS_2_ thin film. Usually, a single layer of TMDC material forms an atomic tri-layer, which consists of two adjacent layers of chalcogen atoms (X) covalently bonded by a layer of transition-metal atoms (M), forming an X-M-X layer configuration. This structure corresponds to two possible structural polytypes: the semiconducting trigonal prismatic phase (1H or 2H phase) and the metallic octahedral prismatic phase (1T) [34,35,36,37]. Figure 2 represents the crystal phases and crystallinity of WS_2_ films of different gas flow rates. It reveals that films deposited at different Ar flow rates exhibit two primary peaks of (101) and (112) orientations. All the films are found at the 3R phase and exhibit the most intense peak at 2θ = 34.9°, corresponding to (101) plane at 5 sccm gas flow rate. When TMDC crystals have more than one atomic sulfur-metal-sulfur layer of the 1H phase bonded by van der Waals (vdW) forces, additional polytypes appear in response to variations in stacking orders. This stacking order is known as Bernal stacking that portraits the rhombohedral phase denominated as the 3R phase. It is found that the XRD peak almost overlaps each other at a higher gas flow rate. Noteworthy that a slight peak shift has been observed for WS_2_ films in this experiment, which may be due to expansion or compression of the lattice caused by stress. Finally, we can conclude that as-grown films are almost pure in nature and did not oxidize during the process of fabrication [38].

Peak width (β) is inversely proportional to crystallite size (L). In this study, the β value is almost the same for all variations. The details of the XRD analysis are given in Appendix B. The average particle size or crystallite size was calculated from the broadening of the (101) peak using the Scherrer equation [39].
L_hkl_ = 0.9λ/(βcos θ)(1)
where L, λ, β, θ are the crystallite size, wavelength, β and θ is the angle between the incident and scattering planes, respectively. Figure 3a represents the crystallite size and growth rate of WS_2_ at different Ar gas flow rates. The crystallite size of the film ranges from 35.3 to 44.8 nm and follows the same pattern with the growth rate. It has been observed that crystallite size increases initially with the increase in Ar gas flow rate but starts decreasing from 6 sccm of gas flow rate and found lowest at 7 sccm Ar flow rate.

Figure 3b represents the film thickness (Appendix A) and growth rate of WS_2_ at different gas flow rates. It has been found that the growth rate increases initially with the increase in gas flow rate and decreases drastically when the gas flow rate increases from 5 sccm flow rate. The highest growth rate has been obtained at 5.69 nm/min at 5 sccm Ar flow rate, and the lowest has been found at 3.37 nm/min at 7 sccm Ar flow rate.

### 3.2. Williamson–Hall (W-H) Analysis

Lattice strains can be developed in the films via scattered grains distribution and/or relocation of the atoms from their reference-lattice positions; however, these phenomena are in turn dependent on the films’ predation conditions, including deposition parameters and sub-sequent annealing conditions. Simply, the lattice strain that developed in the film can be known by estimating “micro-strain”. Scherrer equation neglects the importance of the micro strain (ε), whereas its effects remain present in the diffraction pattern, as this intrinsic strain also produces the broadening in the X-ray profile [40,41]. The contribution of the micro strain to the line broadening of the diffraction peak is defined by Stokes and Wilson and can be calculated as:β_ε_ = 4ε tan θ(2)

The Williamson–Hall equation varies with tan θ only, whereas the Debye–Scherrer equation follows 1/cos θ [42,43]. The addition of Equations (1) and (2) gives the observed broadening (βhkl) assuming the contribution of particle size and strain:β_hkl_ = β_L_ + β_ε_(3)
β_hkl_ = 0.9λ/L cos θ + 4ε tan θ(4)
β_hkl_ cos θ = 0.9λ/L + 4ε sin θ(5)

Williamson–Hall (W-H) plot of as-deposited WS_2_ film at 5 sccm and 7 sccm Ar flow rates are represented in Figure 4 and Figure 5, respectively. W-H plotting consists of a plot of β_hkl_ cos θ versus sin θ, which becomes a straight line. The slope provides the value of micro strain, and the mean particle size can be obtained from the intercept. The highest strain was found 75.52 × 10^−4^ at 6 sccm Ar gas flow rate, while the lowest was observed at 5 sccm gas flow rate, which is 2.36 × 10^−4^. The negative slope of strain that occurs due to the staking fault of atoms indicates the compressive strain [44]. The lower strain value denotes the larger crystallite size.

Figure 6 represents the relationship between crystallite size and strain with respect to Ar flow rate and revealed that 5 sccm Ar flow rate provides a larger crystallite size and lowest strain value for WS_2_ film. This phenomenon is completely in agreement with the literature as discussed above. Geometric parameters obtained from XRD are given in Appendix B (Table A1).

### 3.3. Study on Structural Deformation and Defects

The lattice strains exhibit displacement of atoms from their original lattice positions. It may occur due to high energetic deposition. In addition, a lower value of strain signifies higher crystallinity [40]. Alternatively, the atoms that displace from its reference lattice could act as interstitial atoms and its reference place could act as an unlike vacancy in the crystals. The number of atoms that are displaced from their reference lattice could realize via estimating the dislocation density. Dislocation density can be determined by the following equation [44,45,46,47,48]:Dislocation density; δ = 1/L^2^(6)
where L have their usual significances as mentioned above section. Strain and crystallite size (L) are inversely related. Dislocation density (δ) and crystallites per unit area are represented in Figure 7. It is noteworthy that dislocation density and crystallites per unit area follow the same trend. It is found that dislocation density suddenly rises sharply when films are deposited over 5 sccm Ar gas flow rate. It signifies that imperfections in the crystal lattice, commonly termed as defects that can be either point defect or line defect, increases at a higher Ar flow rate. Moreover, at higher gas flow rates, spikes (discussed in the later part) are being observed. Hence, it can be derived that variation in gas flow rate during RF magnetron sputtering may lead to imperfections in the crystal lattice by changing the crystallographic structure of WS_2_ thin film.

A perfect crystal (Figure 8i) contains the same lattice matrix that has the same unit cell throughout the crystal. The term imperfection or defect is generally used to describe any deviation of the ideally perfect crystal from the periodic arrangement of its constituents. TMDC materials mostly exhibit interface defects due to stacking fault or line defects [37]. Figure 8ii represents the schematic of the stacking defect, and Figure 8iii shows the schematic of the line defect, respectively. The adjacent layers in WS_2_ are associated with the vdW forces. The intensity of vdW forces depends on the layer spacing, which is correlated to the configuration of the stacking. WS_2_ is usually layered as Bernal stacking (AbAb), but deviations from the structure can also be made possible due to the process of synthesis. In contrast, the vacancies of sulfur in the WS_2_ atomic structure forms lines defects, which usually been observed in PVD methods. It is reported that both single and double-line vacancies are observed experimentally for WS_2_ [36,49,50].

### 3.4. Morphological Analysis

In the previous section, it was revealed that as-grown WS_2_ thin films have defects at different scales at their respective crystallographic structure. For visualizing the presence of defects, we have further analyzed its morphological properties through FESEM images. Figure 9 shows the top view of deposited WS_2_ films under different gas flow rates. All the films appear to be non-uniform and porous microstructures (Detail image in Appendix A). As-grown films show lamellar morphology for all the variations of gas flow rate, which is also supported by Regula et al. [50]. Interestingly, spikes are observed on the layered structure of WS_2_. The reason for these spikes might be the stacking deformation or edge defect due to sulfur deficit. We also observe that the number of spikes gradually increases with the increase in gas flow rate.

The average size of observed spikes and grain size with respect to different gas flow rates have been measured by ImageJ software [51], as illustrated in Figure 10. No significant changes have been observed on the average grain size of as-deposited WS_2_ film for different gas flow rates. However, it has been found that the length of spikes gradually decreases with the increase in gas flow rate and is smallest at 5 sccm Ar flow rate. Above 5 sccm gas flow rate, it starts to increase again. This might be due to the loss of S atoms during WS_2_ deposition at higher gas flow rates or due to the sputtering damage of the film [26,52].

Figure 11 depicts the highly magnified FESEM images of WS_2_ thin film deposited on 5 sccm and 7 sccm of Ar gas flow rates. Very low crystallographic deformation has been observed at 5 sccm gas flow rate, and on the other hand, maximum deformation was found at 7 sccm gas flow rate. As at lower gas flow rate exhibits least crystallographic deformation; hence, it can be said that structural defects can be passivated for WS_2_ thin film in PVD method by varying Ar flow rate. The mean free path of a sputtered atom is almost fixed for a system. Sputtered atoms have to go through a number of collisions before reaching the substrate. Increasing working gas flow rate is highly suspected to result in increased energy flux at the substrate surface, which can lead to a higher deposition rate [20]. Again, at a very high gas flow rate (more than an optimized one), the number of atomic collisions is more that results in the higher rate of scattering of sputtered atoms. Thus, too high a gas flow rate reduces the deposition rate and increases the penetration depth of an atom [53,54].

### 3.5. Compositional Analysis by EDX

To investigate and to reconfirm the type of defect of this experiment, we have conducted EDX analysis (Appendix A in Appendix A) by using Area Scan Mode on the film surface. Figure 12 represents the S to W ratio for different Ar flow rates during WS_2_ deposition. It reveals that atomic percentage (at %) of S increases with the increase in gas flow rate and is found highest at 5 sccm flow, which is 1.81:1, and lowest is observed at 6 sccm flow rate, which is 1.44:1. At a very higher gas flow rate, the atomic % of S is much less. It might be because of the scattering effect during deposition of the thin film at a higher gas flow rate. This phenomenon agrees with our findings in the structural properties.

To obtain the best properties from any compound, it is essential to achieve stoichiometry of that compound. From this experiment, it is found that all the as-grown WS_2_ films by RF magneto sputtering are sulfur (S) deficit and far away from stoichiometry. Hence, we can conclude that the deformation or defects in WS_2_ structure is due to the sulfur deficit and can be classified as line defect [55,56].

### 3.6. Electrical Analysis

By examining the electrical properties of a thin film, defect states can be evaluated [55,57]. In this study, it has been revealed that carrier concentrations of as-deposited WS_2_ films are quite high regardless of the gas flow rate variations, and all the films exhibit n-type semiconducting properties. Figure 13 represents the electrical properties of as-grown WS_2_ thin film under different gas flow rates. The highest carrier concentration of 9.31 × 10^19^ cm^−3^ has been observed at 6 sccm gas flow with the resistivity of 2.01 Ω-cm. The highest carrier mobility of 2.42 cm^2^/V.s with a carrier concentration of 3.58 × 10^18^ cm^−3^ have been found at 3 sccm of Ar flow rate. All the electrical properties of the film are given in Appendix B (Table A2).

In the case of semiconductor thin films, carrier concentration and mobility are considered two key factors that control the carrier transport properties. The carrier mobility is mainly affected by impurity scattering during high gas flow rate. It has been observed that carrier mobility decreases gradually with the increase in gas flow rate (Figure 13a). It is considered that the reduction in carrier mobility at a higher argon flow rate is due to an enhanced ionic impurity scattering, which also causes a great number of native defects. On the other hand, the resistivity and carrier concentration of the film gradually increases with the increase in gas flow rate (Figure 13b). The cause is that a higher argon flow rate increases the working pressure near the reaction zone. Consequently, the frequency of the collisions between the ionized atoms and the residual gas also increases. The higher resistivity of the WS_2_ samples produced under the higher gas flow rate is, therefore, due to an increased ion dispersion effect caused by the greater number of native defects. [53,54,58,59]. Hence, a higher gas flow rate influences the yield rate of the sputtered atoms, which has an impact not only on film electrical properties but also on its structural defects. It is already reported that crystal defects have strong effects on the concentration and mobility of the films. Defect-free crystals provide higher carrier transportation property [60,61,62]. Thus, it can be concluded that the 5 sccm Ar gas flow rate effectively tailored the structural defects of WS_2_ thin film to the least, which has been induced due to fabrication limitations. The above analysis clarifies that the film composition and gas flow rate have a crucial role in electronic film properties; certainly, that has a great impact on the performance of thin film solar cells.

### 3.7. Evaluation of Solar Cell Performance

To validate and to pre-investigate the experimental findings, a Solar Cell Capacitance Simulator (SCAPS-1D) is used to oversee the performance of the proposed CdTe/WS_2_ solar cell. A Solar Cell Capacitance Simulator (SCAPS) is a one-dimensional computer simulation software for simulating the alternating current and direct current electrical attributes of thin film heterojunction solar cells. Although it was encouraged to study primarily CdTe and CIGS-based solar cells, SCAPS is currently used to investigate and validate the characteristics of all types of solar cells with different buffer layers as well. SCAPS-1D software essentially operates on two simple semiconductor equations, such as the Poisson equation and the continuity equation of electrons and holes in a steady state. In this paper, we have incorporated the experimental results of WS_2_ as a buffer layer of CdTe/WS_2_ solar cell and compare it with our previously reported literature. For illumination, a regular AM1.5 G spectrum (1000 W/m^2^; T = 300 K) was used. Interface defect layers (IDL) of 10 nm thickness were used to assess the effect of defect densities subsisting on material interfaces. In addition, the neutral type of defect model is used in a simulation where the density of the defect in the active layer has been considered 10^10^ cm^−3^. The thermal velocity of electrons and holes of 1 × 10^7^ cm/s, the Gaussian energy distribution with a characteristic energy of 0.1 eV, have been considered for the model. The schematic structure of the simulated solar cell and the parameters used for this simulation are shown in the Appendix A in Appendix A, Appendix A, respectively. To evaluate and validate WS_2_ as window layer material, initially, CdTe has been chosen as an absorber layer. The reasons for choosing CdTe are because of its availability in nature, comparatively low cost, commonly used in the industry, and most importantly, the matching in energy band. In this simulation, experimental data has been used to obtain the real scenario of the solar cell. Figure 14 shows the IV curve of WS_2_-CdTe solar cell from SCAPS, and the inset shows QE% of the proposed device. From the simulation, it is revealed that WS_2_-CdTe cell has exhibited V_oc_ of 1.04 V, J_sc_ of 26.94 mA/cm^2^, FF of 85.39%, and efficiency of 23.99%. This result super passes our previous experimental findings (1.8% of efficiency) and encourages for commercialization of the device.

## 4. Conclusions

WS_2_ films were deposited at different Ar gas flow rates by sputtering techniques. The deposited films possess a composite structure and exhibit peaks at (101) and (112) planes with a dominant orientation of the (101) plane. All the films are found at the 3R phase, which indicates the formation of a film with Bernal stacking order. The structural analysis reveals the formation of various degrees of defects at a higher gas flow rate, while the EDX analysis suggests that sulfur deficiency could be one of the reasons for the defects at a higher gas flow rate. However, 5 sccm Ar flow rate provides the largest crystallite size of 35.6 nm with the least dislocation density. FESEM images exhibit the presence of defects through visual inspection. All the films are found with non-uniform and porous microstructures. Spikes have also been observed in WS_2_ crystallographic structure, which suggests the existence of native defects in WS_2_ against different gas flow rates. From the electrical analysis, it has been observed that whereas carrier concentration and resistivity of the film increase with the increase in Ar gas flow rate, carrier mobility decreases in this case. The highest carrier concentration of 9.31 × 10^19^ cm^−3^ has been found at 6 sccm Ar flow rate with the resistivity of 2.01 Ω-cm. The highest carrier mobility of 2.42 cm^2^/V.s has been detected at 3 sccm Ar flow rate with the carrier concentration of 3.58 × 10^18^ cm^−3^. It indicates that carrier mobility is mainly affected by impurity scattering during high gas flow rate. The plasma state, therefore, aggravates at a higher gas flow rate and leads to an increased number of ion defects in the film. Hence, it can be said that tailoring of structural defects of WS_2_ thin film is possible by the variation of Ar gas flow rate in the RF magnetron sputtering technique. Finally, taking the experimental data of fabricated WS_2_ at 5 sccm Ar flow, a complete solar cell with a novel structure of FTO/ZnO/WS_2_/CdTe/Ag has been simulated, and 23.99% conversion efficiency was achieved, which encourages the commercial fabrication of the proposed device.

## Figures and Tables

**Figure 1 nanomaterials-11-01635-f001:**
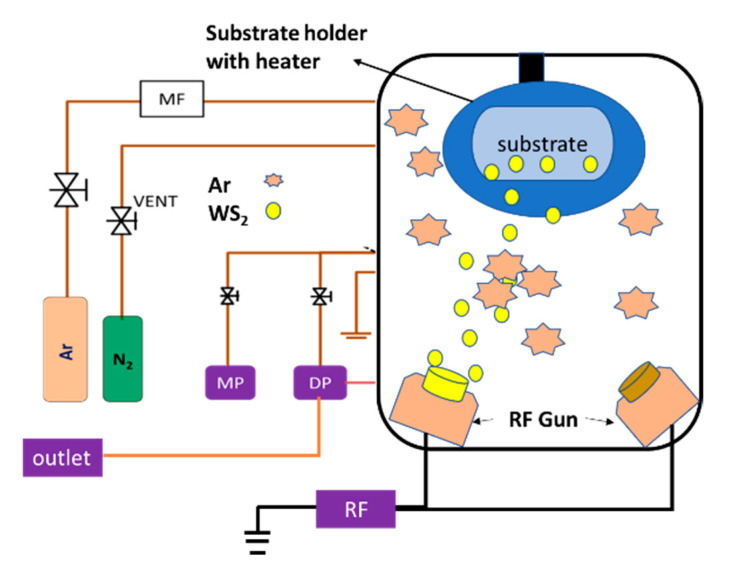
Schematic diagram of RF magnetron sputtering system.

**Figure 2 nanomaterials-11-01635-f002:**
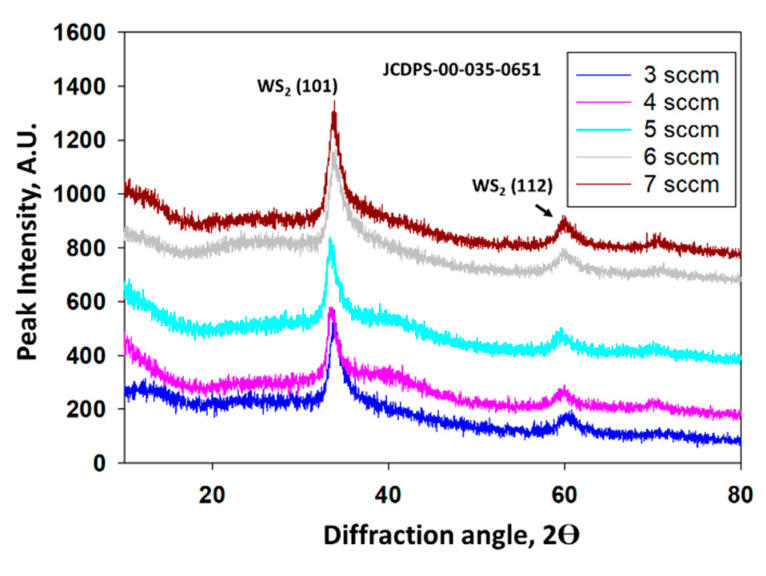
XRD patterns of WS_2_ under different gas flow rates.

**Figure 3 nanomaterials-11-01635-f003:**
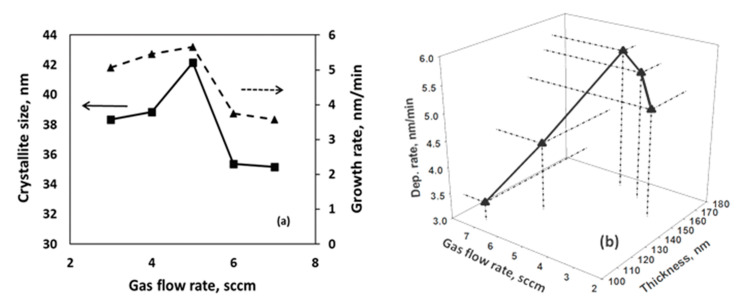
(**a**) Crystallite size and growth rate and (**b**) thickness and deposition rates of as-grown WS_2_ under different Ar gas flow rates.

**Figure 4 nanomaterials-11-01635-f004:**
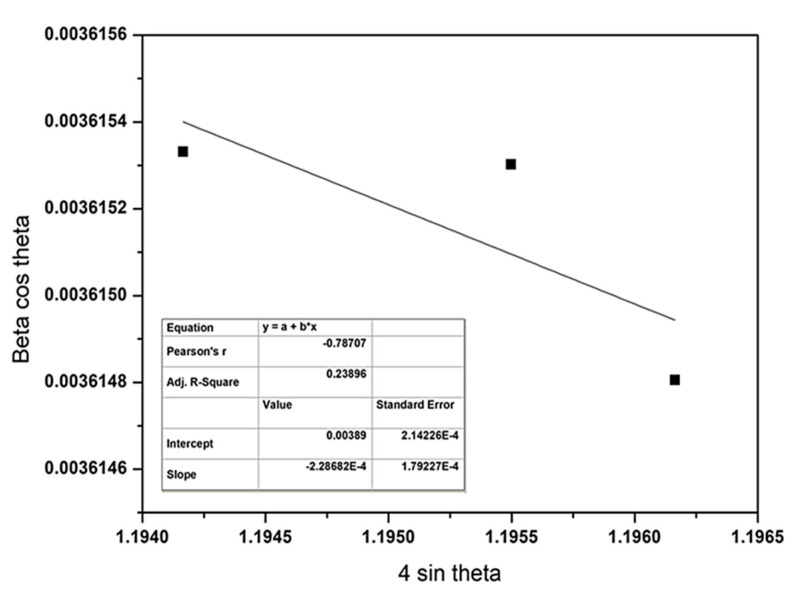
Williamson–Hall plot of as-deposited WS_2_ film at 5 sccm of Ar flow rate.

**Figure 5 nanomaterials-11-01635-f005:**
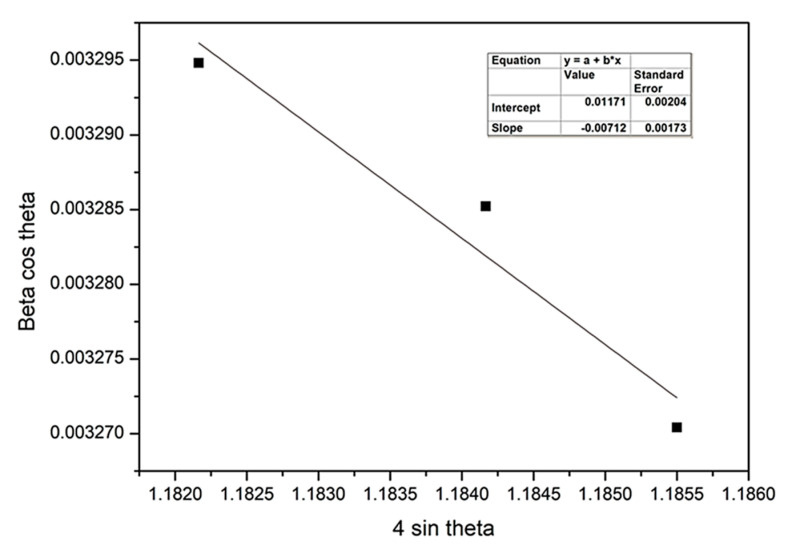
Williamson–Hall plot of as-deposited WS_2_ film at 7 sccm of Ar flow rate.

**Figure 6 nanomaterials-11-01635-f006:**
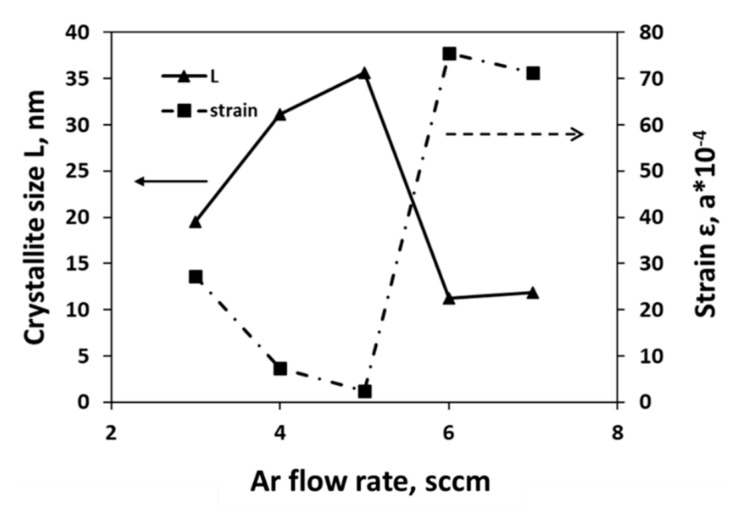
Relation between crystallite size (L) and strain for different gas flow rates of as-deposited WS_2_ films.

**Figure 7 nanomaterials-11-01635-f007:**
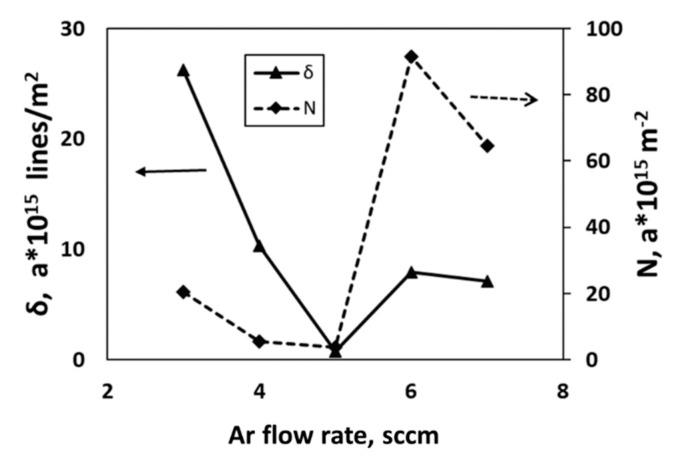
Relation between dislocation density (δ) and crystallites per unit area (N) for different Ar gas flow rates of as-deposited WS_2_ films.

**Figure 8 nanomaterials-11-01635-f008:**
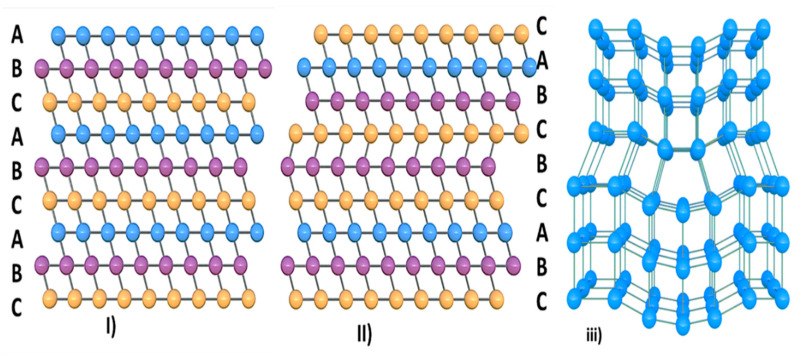
Schematic of defects in TMDC materials: (**i**) perfect crystal; (**ii**) stacking defect; and (**iii**) line defect.

**Figure 9 nanomaterials-11-01635-f009:**
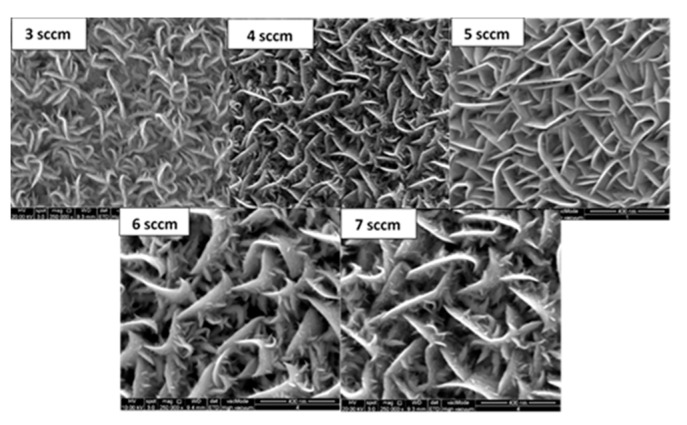
Top view of deposited WS_2_ films under different gas flow rates.

**Figure 10 nanomaterials-11-01635-f010:**
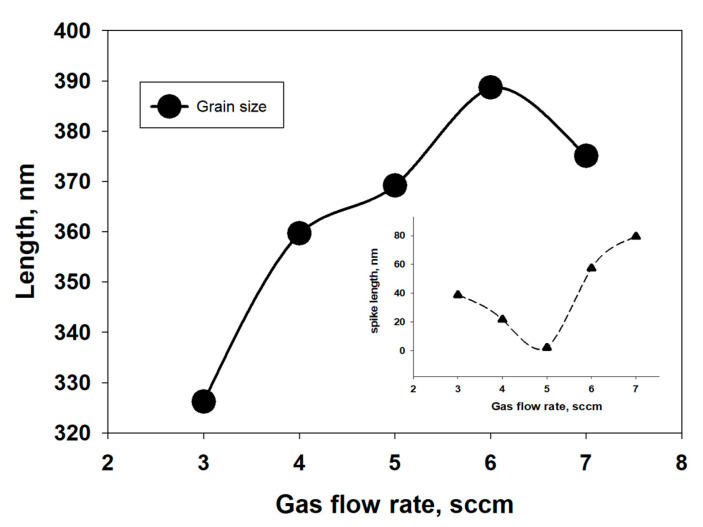
Spike and grain length vs. Ar gas flow rate.

**Figure 11 nanomaterials-11-01635-f011:**
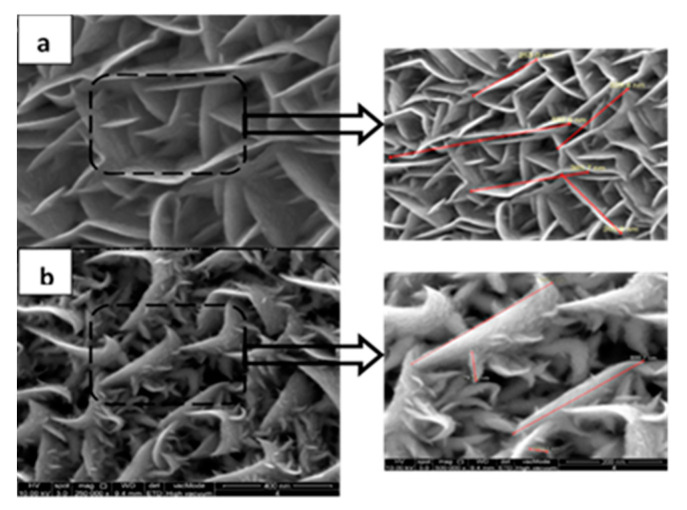
Highly magnified FESEM image of as-grown WS_2_ at different gas flow rates (**a**) 5 sccm and (**b**) 7 sccm.

**Figure 12 nanomaterials-11-01635-f012:**
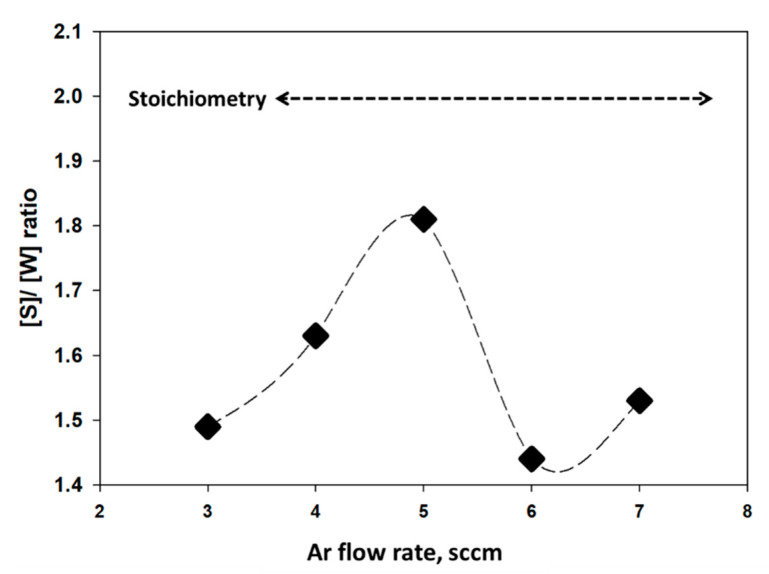
S/W ratio in as-grown WS_2_ under different Ar gas flow rates.

**Figure 13 nanomaterials-11-01635-f013:**
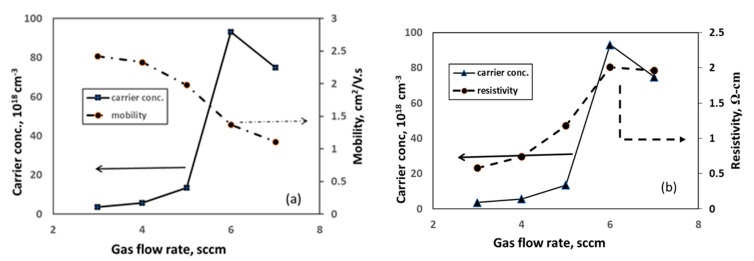
Electrical properties of as-grown WS_2_ thin film under different Ar gas flow rates: (**a**) carrier concentration and mobility (**b**) carrier concentration and resistivity.

**Figure 14 nanomaterials-11-01635-f014:**
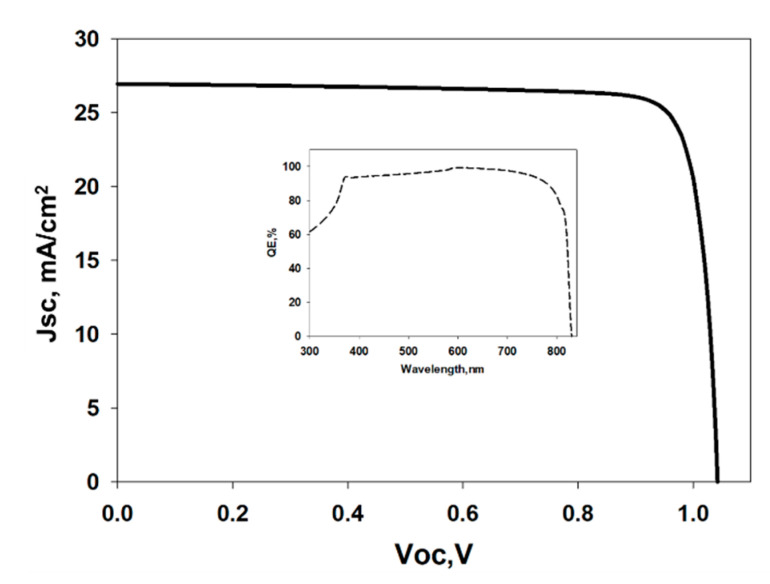
J-V curve of CdTe/WS_2_ solar cell; QE% of CdTe/WS_2_ solar cell at inset.

**Table 1 nanomaterials-11-01635-t001:** WS_2_ fabrication parameters.

Parameters	WS_2_ Fabrication Process
Preheat temperature Preheat time Growth temperature	100 °C 15 min 100 °C
RF power Ar gas flow	50 W 3, 4, 5, 6, 7 sccm
Base pressure	10^−6^ Torr
Operating pressure	1.3~1.8 Pa
Deposition time	30 min

## Data Availability

Data is available upon reasonable request from the corresponding author.

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
