# Peer review of "Impact of Ar Flow Rates on Micro-Structural Properties of WS2 Thin Film by RF Magnetron Sputtering"

_nanomaterials, 2021, doi:10.3390/nano11071635_

Round 1

Reviewer 1 Report

The authors has addressed the comments.

Author Response

Dear Sir,

With due respect, I would like to express my gratitude to you for reviewing my article.

Based on the comments made by reviewers, I have modified/corrected/inserted all the necessary details and highlighted in the manuscript as well.

Thanks.

Dr Md Akhtaruzzaman

Associate Professor

SERI

National University of Malaysia

Reviewer 2 Report

In figure 2, the unit of Y-axis should be (A. U.). In addition, please add the reference (JCDPS) in Figure 2 to confirm the structural characteristics clearly. The main peak is slightly shifted in some case, please explain this.

Author Response

(The authors gave the same response as above.)

Reviewer 3 Report

The manuscript titled “Impact of Ar Flow Rates on Micro-structural Properties of WS2 Thin Film by RF Magnetron Sputtering” is presenting investigations on the effect of deposition conditions (Ar flow) on some properties of WS2 coatings.

I am assuming, from the highlighted parts of the manuscript, that it was reviewed before (not by myself). Consequently, I treated it like a new submission. This paper is not recommended for publication in the Nanomaterials journal, in the present form.

Following are only some of the issues in the manuscript which are to be seriously addressed for accepting its publication in the journal:

  1. The English language and grammar used in the present manuscript is generally poor. There are plenty of instances where mistakes, misspelled words and/or poorly chosen words (ambiguous), and unnecessary repetitions are present. I strongly suggest that the paper should be proofread and double-checked concerning the spelling and phrasing. This version is very difficult to read and to understand.
  2. If the coatings are not stoichiometric, maybe they should be referred as WSx, not WS2.
  3. Several details about the deposition setup are needed: chamber size, argon positioning line, type of magnetron, etc. Are the magnetrons at an angle respective to the substrate holder, like in the schematic? Otherwise, the depositions cannot be reproduced.
  4. The diffraction peaks are clearly shifted, contrary to what the authors are saying. Even in table A1, the peak positions are different. I think that a deconvolution is needed.
  5. I cannot see a connection between the peak-shift and the material being “pure in nature and did not oxidized”. This affirmation should be explained.
  6. Figure 3, I think the caption is incorrect, and it should be rewritten. Figure 3a, add all the increments for the gas flow rate. Grids would be helpful, also.
  7. Why is it that the crystallite size variation has that peak at 5sccm?
  8. Maybe the thickness of the films was obtained from SEM and not EDX data? If so, why not show the cross-section?
  9. Again, why is it that the highest growth rate is observed for 5sccm?
  10. If the films were not prepared in any other way after deposition, the “as deposited” term should be removed.
  11. The SEM images are of poor quality (pixelated).
  12. Explain how the “spikes” average size was obtained. What are the statistics?
  13. “Very low crystallographic deformation has been observed at 5 sccm gas flow rate and on the other hand maximum deformation was found at 7 sccm gas flow rate.” In relation to figure 11. Explanations are needed.
  14. The defects in the coatings could be probably observed by TEM, thus reducing the suppositions.

Author Response

(The authors gave the same response as above.)

Reviewer 4 Report

The paper is devoted to the study of the finest structural features of the WS2 thin films synthesized using magnetron sputtering at different Ar gas flow rates.  The authors have done a very thorough and useful work. Results are presented clear, article is well structured and illustrated.  Hope it will be interesting for many researches working in the field of nanotechnology and recommend it for publication in NANOMATERIALS.

Author Response

(The authors gave the same response as above.)

Round 2

Reviewer 3 Report

My comments were addressed satisfactorily

This manuscript is a resubmission of an earlier submission. The following is a list of the peer review reports and author responses from that submission.

Round 1

Reviewer 1 Report

Major comments:

  1. Can the authors clarify the difference of the working pressure of 10-2 Torr maintained throughout the deposition (line 102) and the operating pressure of 14 mTorr in Table1?
  2. The authors have shown the resistivity and carrier concentration but there is no film thickness reported there? Can the authors provide the film thickness of their sample since they have deposited them for 30min each?
  3. As the authors have mentioned that their WS2 film can be used on solar cell applications, can the authors enlightened where their WS2 film can be applied on the various layer of a solar cell if the film looks non-uniform and porous with “spikes”(line 340)?
  4. Can the authors provide a readable scale for the FESEM image Fig. 9 and Fig. 11?

Minor comments:

  1. Can the authors be more consistent with the use of subscript like WS2 and WS2 and also 100oC and 100°C?
  2. Spotted some grammar mistake and typo, can the authors check again on the grammar, typo like line 89 and the usage of superscript like 10-6Torr (line 101), 10-2Torr (line 102), figure 8 (I) (II) (iii) in the manuscript, RF magneto sputtering (line 425)?

Reviewer 2 Report

The paper investigates the influence of argon gas flow on WS2 film properties. There are severe shortcomings. The paper is not well presented, contains little information regarding the plasma conditions, and provides no explanation for the observed dependencies.

Specific points

Line 16: di sulfide or di-sulfide?

Line 21: W2N ?

Subscripts and superscripts missing, e.g., WS2 in lines 98, 106, 148, 153, 187, 303, table 1, etc., N2 on line 98, beta_hkl on lines 203 and 206, exponents on lines 101, 102, 108, etc.

Line 105 and table 1: changing the gas flow also changes the gas pressure? How much?

Replace Torr by Pa, e.g, on lines 101, 102, table1

Figure 2: peaks appear asymmetric. Why?

Figures 4 and 5: which reflections are plotted? There are only 2 reflections indicated in figure 2. Regarding analysis: strain effects of films deposited by magnetron sputtering and its analysis were investigated by, e.g., Majumdar et al, Coatings 2017, 7, 64; doi:10.3390/coatings7050064

Line 282: what are theta and beta doing here? Delete this sentence, L is already defined before.

Figure 8: what are A, B, C composed of?

Line 373: “as” instead of “an”

Line 374: “is” or “has”?

Line 376: “is smallest at 5 sccm”

Lines 392/404: What do you want to say? Mean free path of an atom is not fixed but depends on species, possible reactions, and kinetic energy. What about ions? Why does it change with gas flow rate if pressure is kept constant (how, see above). How do you know that deposition rate changes? How can an increased gas flow lead to an increase of penetration depths?

Line 404: explain “penetration depth of an atom” (which atom in what?).  

Reviewer 3 Report

The manuscript by Akhtaruzzaman et al. discusses about the characterization of WS2 deposited by magnetron sputtering. This is a topic that is not fairly well studied, and it has considerable interest. The paper presents interesting data, but I mean that it at the present state is shallow to allow publication in Nanomaterials. I offer some comments below that the authors may want to consider for publication in the future:

  1. In general, all the results are disconnected in section with few general considerations, and not considering enough relationship between the different characterizations.
  2. In the section of XRD, authors refer their results to sample at 5 sscm of AR, being all the results quite similar. The discussion should be more general.
  3. Figure 3 is not in the text
  4. It is important to characterize the thickness in every case or Ar flow.
  5. The changes in stoichiometry may due to changes in the cathode, so authors should characterize it by EDX.